# Termination Properties of Transition Rules for Indirect Effects

### Abstract

Indirect effects of agent's actions have traditionally been formalized as condition-effect rules that always fire whenever applicable, after each action taken by the agent. In this work, we investigate a core problem of indirect effects, the possibility of infinitely long sequences of rule firings. Specifically we investigate the termination of rule firings, as well as their confluence, that is, the uniqueness of the state that is ultimately reached. Both problems are PSPACE-complete, and hence far more challenging than what existing literature suggest. To tackle this complexity, we devise practically interesting syntactic and structural restrictions that guarantee polynomial-time termination and confluence tests. Finally, in the context of planning languages that support indirect effects, we propose new implementation technologies.

## 1 Introduction

Actions that an agent takes can have un-anticipated, complex *indirect* effects. Formalizing them as part of the *direct* effects can be impractical, requiring representing them separately. Indirect effects can be formalized as condition-effect rules similarly to agents' actions, but with the requirement that they are fired whenever their condition part is *true* (Kartha and Lifschitz 1994).

Much of the expressive power of indirect effects, and the conceptual and computational complexity that follows from this power, has been earlier ignored. Earlier works either pose strong syntactic restrictions on indirect effects, limiting both their power and the complexity dramatically, or have left the problem unexplored.

The two core questions, which are now addressed in this work, are the possibility of infinitely long sequences of indirect effects, and the impact of different orderings in which the rules for indirect effects are fired. These questions are respectively known as *termination* and *confluence*.

Non-termination is the result of indirect effects generating an infinitely long state sequence, in which the same rules for indirect effects are fired repeatedly.

Confluence is the property of there being a unique terminal state. Conflicting indirect effects may, but do not have to, lead to two or more different terminal states. Indirect effects may override each other's effects, or disable or enable each other, so that the *order* in which they are executed impacts which terminal state is reached in the end. If there is

no unique terminal state, then it is unclear what is a correct implementation of the indirect effects. Questions about confluence (also known as the *Church-Rosser property*) have been researched in connection with different types of rewriting systems (Rosen 1973; Sethi 1974; Keller 1974; Jensen 1980). Our work is the first one to address it explicitly and in full generality for state-space search problems expressed as precondition-effect rules, as used in planning, reasoning about action, and related problems.

We first show that testing for both termination and confluence are PSPACE-complete. This entails that there is no general polynomial-time reduction from planning languages with indirect effects to languages without them, giving a strong justification for introducing indirect effects as an explicit concept in modeling languages for planning and related problems, and motivating at looking at tractable methods to determine confluence and termination. Despite the high worst-case complexity, many practically interesting planning problems have indirect effects that can be effectively analyzed and implemented. We investigate structural properties of rule sets for indirect effects that can be tested in polynomial-time and which yield sufficient conditions for confluence and termination. The methods we propose are based on graphs that represent relations between forced actions, and we obtain tractable termination and confluence tests by limiting to such graphs that are acyclic, or even more strictly, trees or chains.

To make our results as broadly applicable as possible, we consider indirect effects in connection with the most basic model of acting and planning, the classical deterministic planning model. Our results directly apply to more general forms of planning that include the classical planning model.

The structure of the paper is as follows. We start by discussing related work in Section 2. Section 3 formally defines the planning problem that interleaves agent's actions with the execution of all applicable indirect effects. Section 4 defines the *termination* and *confluence* properties, and determines their worst-case complexity. In Section 5 we give computationally easier *sufficient* conditions for confluence and termination. Section 6 proposes two approaches to implement classical planning with indirect effects. Finally, Section 7 reports on experiments with a basic implementation, before we conclude the paper in Section 8 by pointing out future work.

## 2    Related Work

Indirect effects have first been investigated as a part of solutions to the *frame problem* and the *ramification problem* in artificial intelligence (Kartha and Lifschitz 1994). Many works have assumed, for simplicity, that the indirect effects of an action are always mutually consistent, or that conflicting indirect effects mean that an action cannot be taken (Giunchiglia and Lifschitz 1998; Gelfond and Lifschitz 1998).

Planning research has similarly adopted different notion of indirect effects, often called (exogenous) *events*. Petrick and Bacchus (2002) have *update rules*, which are essentially indirect effects. Fox et al. (2005) require that a rule for indirect effects is only fired once. Fox and Long (2006) introduce *event-deterministic* events, which exclude simultaneously applicable conflicting indirect effects, but do not give effective ways to achieve this. A variant of Theorem 1 shows that testing if indirect effects are event-deterministic is PSPACE-complete.[1] None of these works analyze the concept of indirect effects deeper.

Indirect effects in the above works, similarly to ours, are instantaneous. A different model, used in temporal planning, assigns a temporal duration to indirect effects (Gerevini, Saetti, and Serina 2006). Other works consider unpredictable and non-deterministic exogenous events (Iocchi, Nardi, and Rosati 2000; Fritz and McIlraith 2008).

## 3    Preliminaries

Next we define our formal framework. We consider planning with Boolean state variables only.

**Definition 1 (Effects)**  *Let $X$ be the set of state variables. Then the following are* effects *over $X$.*

- *$x$ if $x \in X$*
- *$\neg x$ if $x \in X$*
- *$\phi \triangleright x$ if $\phi$ is a propositional formula over $X$ and $x \in X$*
- *$\phi \triangleright \neg x$ if $\phi$ is a propositional formula over $X$ and $x \in X$*

The notation $\phi \triangleright l$ is for conditional effects: if $\phi$ is true, the literal $l$ will be made true.

**Definition 2 (Actions)**  *Let $X$ be the set of state variables. An action is a pair $(p, e)$ where $p$ is a propositional formula over $X$ for the precondition, and $e$ is a finite set of effects over $X$.*

**Assumption 1**  *For actions $(p, e)$, $\{x, \neg x\} \not\subseteq e$ for all $x \in X$, and if $(\phi \triangleright l) \in e$, then $\bar{l} \notin e$ and for any $(\psi \triangleright \bar{l}) \in e$, the precondition satisfies $p \models \neg(\phi \wedge \psi)$. So no action can make a state variable both true and false at the same time.*

Define complement $\bar{l}$ of literals $l$ by $\overline{x} = \neg x$ and $\overline{\neg x} = x$.

As indirect effects are expressed as condition-effect pairs similar to the agent's actions, we call them *forced actions*, as

---

[1]The definition of *event-deterministic* is stated informally only, and it could alternatively mean actions being *mutex* (Blum and Furst 1997), in which case it is testable in polynomial time.

they must be executed whenever executable. Next we define classical planning extended with forced actions.

**Definition 3 (Problem instance)**  $\langle X, I, A, F, G \rangle$ *is a* problem instance *if*

- *$X$ is a finite set of state variables,*
- *$I \subseteq X$ is the initial state description,*
- *$A$ is a set of actions,*
- *$F$ is a set of actions,*
- *$G$ is a propositional formula over $X$ for the goal states.*

Here $F$ are the forced actions which must be fired whenever their precondition is true. The initial state description $I$ denotes the unique state $s_0$ such that $s_0 \models x$ for all $x \in I$ and $s_0 \models \neg x$ for all $x \in X \backslash I$.

**Assumption 2**  *There are no forced actions applicable in the initial state.*

This assumption is just to simplify the presentation, to avoid separately talking about confluence and termination when forced actions are being triggered before any actions are taken. One could view the facts in the initial states as the unconditional effects of a special initial state action, so there is no loss of generality in this assumption.

**Assumption 3**  *Every forced action makes its own precondition immediately false, so that the forced action cannot be fired multiple times in a sequence without other actions in between. This can be often achieved with an effect literal $l$ such that $l \models p$ where $p$ is the precondition.*

There are other ways to prevent infinite firings of a single action. First, an action could be fired only when its precondition *becomes* true. Second, an action could be fired only when it actually changes the value of at least one state variable (but this would prevent multiple firings only for actions that always have the same effects). But we do not investigate this further.

**Definition 4 (Successor state)**  $s' = \text{exec}_a(s)$ *is the* successor state *of $s$ with respect to action $a = (p, e)$ such that*

- *$s' \models x$ if $x \in e$ or $(\phi \triangleright x) \in e$ and $s \models \phi$,*
- *$s' \models \neg x$ if $\neg x \in e$ or $(\phi \triangleright \neg x) \in e$ and $s \models \phi$,*
- *$s(x) = s'(x)$ if $x \notin e$, $\neg x \notin e$, $s \not\models \phi$ for any $(\phi \triangleright x) \in e$ and $(\phi \triangleright \neg x) \in e$.*

*The successor state is defined if $s \models p$.*

We define $\text{exec}_{a_1, \ldots, a_n}(s) = \text{exec}_{a_n}(\cdots \text{exec}_{a_1}(s) \cdots)$.

**Definition 5 (Executions)**  *An* execution *of (forced) actions in state $s_0$ is a sequence $a_1, \ldots, a_n$ of action and a sequence $s_0, s_1, \ldots, s_n$ of states such that $s_0 = s$ and for all $i \in \{1, \ldots, n\}$, $s_{i-1} \models p_i$ (where $a_i = (p_i, e_i)$) and $s_i = \text{exec}_{a_i}(s_{i-1})$.*

The execution of an agent's action is followed by the execution of all applicable forced actions.

**Definition 6 (Forced executions)** *For* $\langle X, I, A, F, G \rangle$ *a forced execution in state $s$ is any sequence $a_1, \ldots, a_n$ of actions with $a_i \in F$ for all $i \in \{1, \ldots, n\}$ such that $a_1, \ldots, a_n$ is an execution (as in Definition 5) and $\mathrm{exec}_{a_1, \ldots, a_n}(s) \not\models p$ for all $(p, e) \in F$.*

So a forced execution is an execution of forced actions that ends in a state with no executable forced actions.

**Definition 7 (Reachable states)** *We define reachable states for problem instances $\langle X, I, A, F, G \rangle$ recursively.*

- *The initial state $s_0$ for $I$ is a reachable state.*
- *$s'$ is a reachable state if*
  - *$s$ is a reachable state,*
  - *$a \in A$ is an action,*
  - *$a_1, \ldots, a_n$ is a forced execution in $\mathrm{exec}_a(s)$, and*
  - *$s' = \mathrm{exec}_{a, a_1, \ldots, a_n}(s)$.*

So a state is reachable if it can be reached by a sequence of actions from $A$ interleaved with maximal sequences of actions from $F$.

## 4 Properties: Termination and Confluence

Some sets of forced actions have infinitely long executions.

**Example 1** *Consider forced actions $(a, \{\neg a\})$ and $(\neg a, \{a\})$. Starting from any state, these two actions have one execution which is infinitely long.*

So the question of *termination* of forced actions arises: are all executions of forced actions finite?

**Definition 8 (Termination)** *Forced actions in a problem instance are* terminating *if every forced execution for all states $s$ that are reachable in $\langle X, I, A, F, G \rangle$ is finite.*

Often one wishes that the state reached with the forced actions is unique, the ordering in which the actions have been tried is therefore irrelevant, and therefore the implementation of the forced actions can use any ordering with no concerns about it having an impact on the terminal state.

**Example 2** *Consider forced actions $(a, \{\neg a, b\})$ and $(a, \{\neg a, c\})$ and a state $s$ such that $s \models a \wedge \neg b \wedge \neg c$. There are two forced executions of the actions, each consisting of one forced action only, leading to two different states.*

The property we are after is known as the *confluence* or the *Church-Rosser* property (Rosen 1973; Sethi 1974), and shows up prominently for example in the context of the $\lambda$-calculus (Barendregt 1984) and other similar systems in which tree-like syntactic expressions are transformed by a sequence of rewriting steps. We adapt this concept to transition systems and planning.

**Definition 9 (Confluence)** *Forced actions in a problem instance are* confluent *if $\mathrm{exec}_{\sigma_1}(s) = \mathrm{exec}_{\sigma_2}(s)$ for all forced executions $\sigma_1$ and $\sigma_2$ and for all states $s$ that are reachable in $\langle X, I, A, F, G \rangle$.*

**Theorem 1** *Testing for confluence of forced actions is PSPACE-complete.*

*Proof:* Idea: Construct actions for simulating deterministic PSPACE Turing machines (Bylander 1994), with two additional actions, one going from the initial state to a special end state (the short cut), and another going from accepting state to the same special end state. The short-cut action is always applicable. If the Turing machine does not accept, then the execution ends in a non-accepting state different from the one reached by the short-cut action. Hence the forced actions are confluent if and only if the Turing machine has an accepting execution. This shows PSPACE-hardness.

PSPACE membership is by finding a reachable state from which two different terminal states are reachable with forced actions. This is by nested calls to the standard PSPACE reachability test. $\square$

**Theorem 2** *Testing for termination is PSPACE-complete.*

*Proof:* Idea: Construct actions for simulating PSPACE Turing machines (Bylander 1994) with additional two actions that can be repeated indefinitely (similarly to Example 1) if reaching a non-accepting terminal state. Then the Turing machine accepts if and only if the rules are terminating.

PSPACE-membership is by finding a reachable state from which a sequence of forced actions form a cycle in the state space. This is by nested calls to the standard PSPACE reachability test. $\square$

## 5 Sufficient Conditions for Confluence

We discuss sufficient conditions for confluence that can be tested in polynomial time. The first condition is trivial and familiar from earlier works, and is given only as a starting point for the more broadly applicable conditions we develop.

We first formalize four binary relations $R_c$, $R_e$, $R_d$, and $R_i$, and then derive sufficient conditions for confluence. These relations have the following meanings.

$R_c$ the two actions have *conflicting* effects, that is, one can make some state variable true, and the other action can make it false.

$R_e$ One action can *enable* another action by making the latter's precondition true.

$R_d$ One action can *disable* another action by making the latter's precondition false.

$R_i$ One action can *impact* the effects of another action by changing the truth-value of one of the conditions $\phi$ in a conditional effect $\phi \triangleright l$ of the latter.

Next we define the relations formally.

**Definition 10 (Relation $R_c$)** *The relation $R_c \subseteq (A \cup F)^2$ is defined by $a_0 R_c a_1$ whenever $a_0 \neq a_1$ and $\mathrm{atEffs}(a_0) \cap \{\bar{l} \mid l \in \mathrm{atEffs}(a_1)\} \neq \emptyset$.*

Here $\mathrm{atEffs}(a)$ denotes the atomic effects of an action $(p, e)$, defined as $\{x | x \in e\} \cup \{\neg x | \neg x \in e\} \cup \{x | (\phi \triangleright x) \in e\} \cup \{\neg x | (\phi \triangleright \neg x) \in e\}$.

The relation for *enabling* is based on a syntactic condition that refers to the effects of the first action and the precondition of the second.

**Definition 11 (Relation $R_e$)** $a_1 R_e a_2$ *iff* $a_1 \neq a_2$ *and*

1. *there is* $x \in \mathrm{atEffs}(a_1)$ *that occurs positively in the precondition of* $a_2$, *or*
2. *there is* $\neg x \in \mathrm{atEffs}(a_1)$ *that occurs negatively in the precondition of* $a_2$.

The *disabling* relation is defined similarly.

**Definition 12 (Relation $R_d$)** $a_1 R_d a_2$ *iff* $a_1 \neq a_2$ *and*

1. *there is* $x \in \mathrm{atEffs}(a_1)$ *that occurs negatively in the precondition of* $a_2$, *or*
2. *there is* $\neg x \in \mathrm{atEffs}(a_1)$ *that occurs positively in the precondition of* $a_2$.

Additionally, in the presence of conditional effects $\phi \triangleright l$, we need to model the way an action may impact what effects another action actually has. The impact relation indicates whether it is possible that an action changes the effects another action can have.

**Definition 13 (Relation $R_i$)** $a_1 R_i a_2$ *iff* $a_1 \neq a_2$ *and* $x \in \mathrm{atEffs}(a_1)$ *or* $\neg x \in \mathrm{atEffs}(a_1)$, *and* $x$ *occurs in* $\psi$ *for some effect* $\psi \triangleright l$ *of* $a_2$.

The simplest confluence test is the following.

**Theorem 3** *For a problem instance* $\langle X, I, A, F, G \rangle$, *the forced actions are confluent if*

1. $R_c \cap (F \times F) = \emptyset$,
2. $R_d \cap (F \times F) = \emptyset$, *and*
3. $R_i \cap (F \times F) = \emptyset$.

*Proof:* Idea: The conditions guarantee that no forced action that is applicable in $s$ or in any state $s$ reached with forced actions becomes inapplicable after firing other forced actions, and the effects of no forced action are overwritten by other forced actions, and the conditional effects of no action depend on whether other actions are fired before or after it. Hence the same actions are fired and the same terminal state is reached no matter what order the actions are fired in.

Proof sketch: Assume that there are two executions that lead to different terminal states. Since none of the actions override the effects of other actions and since the conditions of the conditional effects have the same values in both, the terminal states can only differ because one execution includes at least one action not included in the other.

Note that because of the requirement on $R_i$, the formulas $\phi$ in conditional effects $\phi \triangleright l$ are not changed by any of the forced actions, and hence they are evaluated in the state where the application of the forced actions begins in.

Let $a_1, \dots, a_n$ and $a'_1, \dots, a'_{n'}$ be the two executions, and by symmetry we assume that the first execution includes at least one action not included in the other. Let $a_i$ be the first such an action in that execution. Hence $\{a_1, \dots, a_{i-1}\} \subseteq \{a'_1, \dots, a'_{n'}\}$. Let $j \in \{1, \dots, n'\}$ be such that $a'_j = a_{i-1}$. Since the effects of $a_1, \dots, a_{i-1}$ are not overridden by any action in the second execution, nor are any preconditions of any actions made false, the precondition of $a_i$ must be true right after $a'_j$ in the second execution. And later actions $a'_k, k > j$ will not make the precondition of $a_i$ false. Hence $a_i$ would continue to be applicable for the rest of the execution, and would have to be included in it. This contradicts the assumption that $a_i$ is not part of the second execution. Therefore it is not possible that there are two executions that have different actions. $\square$

These conditions are essentially what is required for actions at the same level in GraphPlan's plans (Blum and Furst 1997) and also in the planning as satisfiability approach (Kautz and Selman 1996). The conditions in Theorem 3 are not necessary for confluence.

**Example 3** *Consider actions* $(a, \{\neg b, c\}), (a, \{b, d\}), (c \wedge d, \{e, b\})$. *Since the first two have conflicting effects, the conditions of Theorem 3 are not satisfied. Nevertheless, the rules are confluent, always leading to the same terminal state. The conflict on* $b$ *between the first two actions is not important, as* $b$ *will always be made true by the last action.*

Next we look at more powerful polynomial time tests for confluence. The first issue with Theorem 3 is that it looks at the set of forced actions as a whole, even though it would be sufficient to limit to only those forced actions that are fired when one of the agent's actions is executed.

We denote the transitive closure of a relation $R$ by $R+$.

**Definition 14 (Enabled Forced Actions)** *Let* $A$ *be a set of actions,* $F$ *a set of forced actions, and* $a$ *an action. Then* $F_a = \{a_1 \in F | (a, a_1) \in (R_e \cap ((A \cup F) \times F))+\}$ *is the set of* enabled forced actions *for* $a$.

The enabled actions is an *over-approximation* of the set of actions that possibly become executable after taking $a$ and some (possibly empty) sequence of actions.

**Theorem 4** *Let* $\Pi = \langle X, I, A, F, G \rangle$ *be a problem instance.* $\Pi$ *is confluent if for every* $a \in A$, *the set of enabled forced actions for* $a$ *has no pair of actions related by* $R_c$, $R_d$, *or* $R_i$.

*Proof:* Like the proof of Theorem 3, restricted to the enabled forced actions for $a$. $\square$

The requirement of Theorem 4 that no two forced actions conflict is sometimes too restrictive. Next we present a more refined condition which is sufficient to determine confluence for a broader class of problems, including one of our sample problems that we will discuss later.

We identify a broader classes of enabling relations that yields confluent planning problems even when actions disable other actions or have conflicting effects. Two actions $a_0$

and $a_1$ respectively with conflicting effects $\neg x$ and $x$ in general violate confluence because there could be two different executions in which these two actions are taken in opposite orders, either making $x$ true first and then making it false, or vice versa, leading to different terminal states.

But, we could allow two actions to have conflicting effects if they are always executed in the same order, and other actions cannot impact their execution so that the conflicting effects could play out in two different ways.

**Definition 15 (Enabling Graph)** *An* enabling graph *for an action $a$ is a graph $G_a = \langle G, E \rangle$ where $G = \{a\} \cup F_a$ for the set of enabled forced actions $F_a$ for the action $a$, and $E = R_e \cap ((\{a\} \cup F_a) \times (\{a\} \cup F_a))$.*

Cycles in enabling graphs are a potential indication of non-termination of forced actions. Enabling graphs in problems such as the one in Example 1 are clearly cyclic. For acyclic enabling graphs the executions of forced actions are finite, but can be exponentially long.

**Example 4** *Consider the following forced actions for all $i \in \{1, \ldots, n\}$.*

$$a_i = (x_i, \{x_{i+1}, x'_{i+1}, \neg x_i\})$$
$$a'_i = (x'_i, \{x_{i+1}, x'_{i+1}, \neg x'_i\})$$

*For an action $a$ with effects $x_1$ and $x'_1$, the graph $G_a$ is acyclic.*

*All of these forced actions are executed after $x_1$ and $x'_1$ become true, but there are several possible executions, and on some of them some actions are executed an exponential number of times.*

*After both $a_{n-1}$ and $a'_{n-1}$ one can execute $a_n, a'_n$. Similarly, after both $a_{n-2}$ and $a'_{n-2}$ one can execute $a_{n-1}, a_n, a'_n, a'_{n-1}, a_n, a'_n$. As $n$ increases, the length of executions of this form increases exponentially as $2^{n+1} - 2$.*

Acyclicity of $G_a$ only guarantees boundedness of executions, but does not guarantee the uniqueness of the terminal state, as different paths in the graphs may be interleaved to executions in different ways.

The same holds also if $G_a$ is a tree: even if there are only two branches in a tree-formed $G_a$ (and the executions are only polynomially long), the branches can be interleaved in an exponential number of different ways, and they could lead to an exponential number of different terminal states. So, confluence requires either even stronger restrictions on $G_a$, like it being a chain, or other additional restrictions.

Next we focus on graphs that are trees, and restrict the properties of forced actions in different branches of the tree so that confluence is achieved. Note that neither acyclicity nor being a tree is a necessary condition for confluence or termination, and we are here interested in practically useful sufficient conditions for these properties, especially ones that can be tested in low-polynomial time.

To guarantee confluence we require that any variation in the order in which the forced actions are executed will never impact which terminal state is reached. In an extreme case,
the tree consists of a single branch, a totally ordered sequence of forced actions, which can only be executed in this particular order. Hence confluence trivially follows. If the tree has several branches, then none of the branches are allowed to impact how the execution in the other branches proceeds, and hence interleaving the different branches in arbitrary ways still always leads to the same terminal state. This is achieved simply by requiring that there is no inter-branch interaction corresponding to three of the relations we have defined, $R_d$ for disabling, $R_i$ for changing the conditions of conditional effects, and $R_c$ for having conflicting effects.

**Theorem 5** *Let $\Pi = \langle X, I, A, F, G \rangle$ be a problem instance. $\Pi$ is confluent if the enabling graph $G_a$ of every $a \in A$ satisfies the following.*

1. *$G_a$ is a tree, and*
2. *for every $(a_0, a_1) \in (F \times F) \cap (R_d \cup R_i \cup R_c)$, there is a directed path from $a_0$ to $a_1$ or from $a_1$ to $a_0$ in $G_a$.*

*Proof:* Idea: The different maximal paths in the tree do not interfere with each other (except of course they may share parts of their initial segment), and attempting execution of any total ordering of the partial order represented by the tree – ignoring actions with a false precondition – leads to executing exactly the same set of actions, with the same terminal state for every execution.

Let $s$ be any reachable state for $\Pi$ and let $a \in A$ be an action applicable in $s$. Let $\sigma_1 = a_1^1, \ldots, a_n^1$ and $\sigma_2 = a_1^2, \ldots, a_m^2$ be sequences of forced actions so that $s_0^1, \ldots, s_n^1$ and $s_0^2, \ldots, s_m^2$ are their respective executions starting with $s_0^1 = s = s_0^2$, and the sequences are maximal in the sense that no forced actions are applicable in $s_n^1$ and $s_m^2$.

We will next show that $\{a_1^1, \ldots, a_n^1\} = \{a_1^2, \ldots, a_m^2\}$ so that exactly the same forced actions are executed after taking action $a$, and that $s_n^1 = s_m^2$ so that the resulting state is the same in both cases.

To show that the same actions are in both, we assume that is an action $a_i^1$ that does not appear in $\sigma_2$, and choose such action with the least $i$. So forced actions $a_1^1, \ldots, a_{i-1}^1$ occur also in $\sigma_2$, including those on the path from the root action $a$ to $a_i^1$. These latter actions cannot interfere with any action in $\sigma_2$, because there is not inter-branch interaction of the forced actions, and hence after the last of those actions the precondition of $a_i^1$ is true. But now none of the forced actions following $a_i^1$ in $G_a$ can be in $\sigma_2$ (as only $a_i^1$ can make them applicable) and therefore cannot falsify the precondition of $a_i^1$, and also forced actions in other branches in $G_a$ cannot falsify the precondition of $a_i^1$, so $a_i^1$ must be included in $\sigma_2$ as well, which is a contradiction with the assumption that there is a forced action in $\sigma_1$ that does not occur in $\sigma_2$. Hence all forced actions in $\sigma_1$ must also be in $\sigma_2$. By symmetry, same holds for $\sigma_2$ and $\sigma_1$, so hence their forced actions coincide.

It remains to show that both $\sigma_1$ and $\sigma_2$ lead to the same terminal state.

We first prove an auxiliary result by induction on the distance of the (forced) actions from the root action $a$ in $G_a$.

Induction hypothesis: For any (forced) action with distance $i$ from the root of $G_a$ that occurs in $\sigma_1$ and $\sigma_2$, in both execution $\sigma_1$ and in $\sigma_2$ the action has the same (conditional and unconditional effects).

Base case $i = 0$: This action is the root action $a$, being executed in the same state $s$ for both $\sigma_1$ and $\sigma_2$, and hence has the same effects in both cases.

Inductive case $i \geq 1$: Let $a'$ be an action in $\sigma_1$ and $\sigma_2$ with distance $i$ from $a$. All (forced) actions that change state variables in the conditions $c$ of conditional effects $c \triangleright e$ of $a'$ before the execution of $a'$ in $\sigma_1$ and $\sigma_2$ have distance $< i$, and hence have exactly the same conditional and unconditional effects in both $\sigma_1$ and $\sigma_2$. Hence the effects of $a'$ are the same in both executions.

Finally, for every state variable $x$ and every pair of actions that change $x$ in $\sigma_1$ and $\sigma_2$, these two actions have the same ordering in both $\sigma_1$ and $\sigma_2$, as that ordering is determined by $G_a$ (actions that have conflicting effects cannot be in unordered in $G_a$, that is, reside in different branches of the tree $G_a$.)

Hence the final value of $x$ is the same in both executions $\sigma_1$ and $\sigma_2$, and hence the terminal states are the same. $\quad\square$

# 6 Implementation

Often it can be guaranteed that taking an action can only trigger the firing of a small number of forced actions (for example by using the concept of *enabled forced action* in Definition 14). In these cases it is possible to implement the forced actions by reducing the whole planning problem to a conventional classical planning problem. If there is only one forced action that could become executable after an agent's chosen action, then simply compose them together, with the precondition $p$ of the forced action first moved to the condition part of its effect $\phi \triangleright l$ to obtain $(p \land \phi) \triangleright l$. However, compiling the potentially applicable forced actions into the enabling regular actions is infeasible if there are very many of them. We have not experimented with this option further.

Next we consider general implementations in two main frameworks, heuristic state space search (Bonet and Geffner 2000) and the planning as satisfiability approach (Kautz and Selman 1996). An underlying assumption in both cases is confluence (Section 4). If a rule set does not have the confluence property, the results of the rule firings are not uniquely determined by the starting state, and will be affected by the ordering in which forced actions are considered. In the state-space search setting the most natural implementations would follow an arbitrary ordering. SAT-based methods could be implemented so that all possible firing orderings would be covered, so that a planner would return a plan whenever a plan exists for at least one firing ordering for every instance of firings, but we will not look at this in more detail.

## 6.1 Heuristic State Space Search

Implementation with standard state-space search algorithms in planners such as HSP and FF (Bonet and Geffner 2000; Hoffmann and Nebel 2001) is straightforward. Followed by a regular action, all applicable forced actions are applied until none are applicable. Assuming confluence, the resulting state is unique.

For computing many well-known heuristics, a straightforward implementation is to treat forced actions and regular actions alike, so not requiring any modification in the heuristic at all. An intuitively plausible modification would be to treat forced actions as having cost 0, but we will see in Section 7 that sometimes this does not work well.

## 6.2 Planning as Satisfiability

Extending encodings of planning as satisfiability (Kautz and Selman 1996) with forced actions is slightly more complicated. We need to force the execution of all forced actions after every regular action. The principles the encoding implements are the following.

1. At least one forced action must be taken whenever the precondition of at least one forced action is true.

2. An ordinary (non-forced) action can be taken only if no forced action is applicable at the same step.

The first requires firing all applicable forced actions, in one or more steps of a *parallel plan* (Kautz and Selman 1996). The second means that agent's actions can only be considered after all applicable forced actions have been fired. Next we give an implementation of these principles.

Let there be $n$ forced actions, and let $p_1, \ldots, p_n$ be their respective preconditions and let $f_1@t, \ldots, f_n@t$ be the atomic propositions indicating whether they are executed at time point $t$. Let $a_1@t, \ldots, a_m@t$ be the atomic propositions indicating whether the agent actions are executed.

Requirement 1 is encoded as follows.

$$p_i@t \to F@t \text{ for all } i \in \{1, \ldots, n\} \tag{1}$$
$$F@t \to (f_1@t \lor \cdots \lor f_n@t) \tag{2}$$

Requirement 2 is encoded as follows.

$$a_i@t \to \neg F@t \text{ for all } i \in \{1, \ldots, m\} \tag{3}$$

Here $F@t$ is true if at time step $t$ there are forced actions.

This is the most basic encoding. We can strengthen the encoding so that forced actions following one regular action are never unnecessarily spread to more time points than what is necessary. The additional condition that handles this is as follows.

1b. If the precondition of the forced action is true and no forced action is executed that either has a conflicting effect or that is mutex with the forced action, then the forced action must be taken.

This could further be strengthened by removing ambiguity about which maximal consistent set of forced actions is taken by adding "no forced action with a lower index", so that in conflict situations always the lower indexed forced action wins. This condition is encoded as follows.

$$p_i@t \to F@t \text{ for all } i \in \{1, \ldots, n\} \tag{4}$$
$$\left( p_i@t \land \neg \bigvee_{j \in C_i} f_j@t \right) \to f_i@t \tag{5}$$

Here $C_i$ is the set of lower-indexed actions that have an effect that conflicts with the effect of forced action $i$ or that is mutex with it, in the sense of Graphplan parallelism (Blum and Furst 1997; Kautz and Selman 1996).

## 7 Experiments

Following Section 6.2, we have implemented forced actions in FF (Hoffmann and Nebel 2001), a well-known planner often used for benchmarking, as well as a basic SAT-based planner. Then we have run experiments, demonstrating two uses of forced actions, viewing the distinction between regular and forced actions as *control knowledge*, and using forced actions to describe complex indirect effects of the agent's actions.

In our first experiment, we reformulated the well-known logistics domain with control knowledge, and then evaluated it against the original version without control knowledge.

In the second experiment we demonstrate the use of forced actions in expressing indirect effects that would be difficult to encapsulate in regular actions. For this, we introduce a new domain derived from the well-known Game of Life (Gardner 1970) cellular automaton.

All experiments were run on Intel Xeon E5-2680 CPUs, with 8 GB of memory and a time limit of 30 minutes.

### 7.1 Control Knowledge for Logistics

Forced actions can also be viewed as a form of control knowledge. Consider the standard logistics domain, with airplanes and trucks for transporting packages. Some of the actions can be viewed as forced actions which should be taken whenever it is possible.

- A package at its target city but not at its target location must be loaded into a truck.
- A package not in its target city must be loaded into a truck if it is not at the airport.
- A package destined to a different city must be unloaded from a truck at the airport.
- A package in its target city must be unloaded from the airplane.
- A package in its target location must be unloaded from a truck.

These forced actions refer to the target cities and locations of the packages, which must be accordingly modeled.

The only actions left to the agent to decide about are the move actions for trucks and airplanes, and whether to load a package to an airplane. Loading a package to an arbitrary airplane may risk losing optimality of a plan.

Note that these forced actions mostly satisfy the conditions of Theorem 3, except for immediate loads after unloads, as in unloading a package from a truck and then loading it into an airplane. The more relaxed conditions of Theorem 5 in this case show that confluence indeed holds.

Our experiment uses 200 instances from the year 2000 planning competition Logistics domain, both in the original version, and a revised version with many of the actions turned to forced actions as described above.

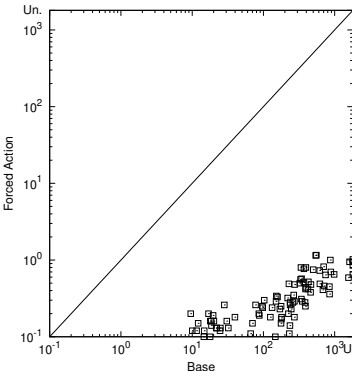

Figure 1: FFF1 and FF runtimes on Logistics

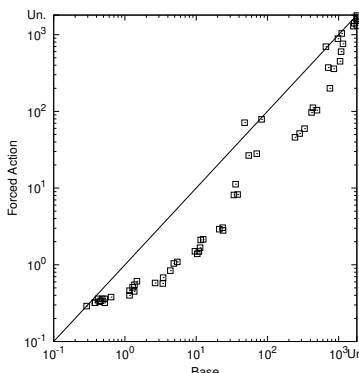

Figure 2: SAT runtimes on Logistics

We first implemented FF with forced actions (which we call FFF0) according to Section 6.1, so that the heuristic computation considers the forced actions as having cost 0. The results turned out to be underwhelming, with FFF0 often not solving Logistics any better. It turned out that forced actions having cost 0 turns the heuristic uninformative and makes the search quite blind. Then we modified the implementation so that forced actions are considered by the heuristic exactly like the rest of the actions, obtaining FFF1. Figure 1 gives the runtimes of FF on the original instances and FFF1 with the same instances enhanced with forced actions. Using forced actions as control knowledge decimates the runtimes also for the very large Logistics instances that FF would otherwise need tens of minutes to solve.

For the SAT approach, we used a basic SAT-based planner with a standard parallel encoding (Kautz and Selman 1996) as a baseline. Then we extended it, as discussed in Section 6.2, to support forced actions (the most basic encoding only), and ran it with the same Logistics problems, with the results given in Figure 2. The experiment used the state-of-the-art KisSAT SAT solver. While the results are not as impressive as with FFF1, substantial runtime improvement is still obtained, often by one order of magnitude.

The results show that control knowledge in the form of forced actions can considerably improve performance when some of the actions can be considered *obligatory* when-

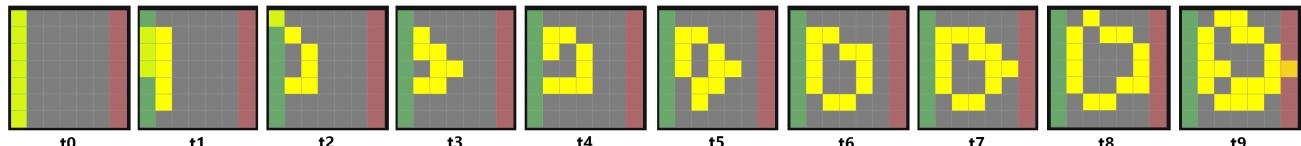

Figure 3: Our Game of Life problem in a $7 \times 7$ grid. It takes ten time steps to reach the destination area (rightmost column) from the source area (leftmost column). The agent sets cells in the source area in the first three steps.

ever they are applicable. Although Logistics is a simple domain and is solved by many planners very efficiently, the reduction in the number of decision points, due to the smaller number of regular actions, still leads to a performance improvement. In some other domains these improvements could be still far bigger.

## 7.2 Game of Life

Game of Life (Gardner 1970) is a cellular automaton. The cells in the grid are either live or dead. Given the current state, the following three rules determine the next state: Any cell with three live neighbors will be alive. Any live cell with two live neighbors will be alive. All other cells will be dead.

For our second experiment, we based a planning problem on Game of Life. We partition a finite grid into three parts: source area, destination area, and the rest. The goal is to make at least one cell alive in the destination area by controlling the cells in the source area, and otherwise allowing the automaton run according to its usual rules. Figure 3 demonstrates an example of this problem with its solution.

We implemented this problem with agent's actions controlling the source area, and with the cellular automaton rules implemented as forced actions. We use two copies of the grid, the old and the new, so that each cell's new state only depends on the cells' old values, with the new grid becoming the next new for the next stage of cell evolution. This problem is confluent, as the forced actions are used to determine the unique next state of the grid evolution.

We generated ten problem instances for grid sizes $3 \times 3$ until $12 \times 12$. The leftmost column is the source area, and the rightmost column is the destination area. The results of this experiment are shown in Figure 4. There is no baseline version of this benchmark, as expressing it in standard modeling languages is impractical. Finding cellular automata states satisfying complex criteria is computationally hard, so the runtimes in this problem unsurprisingly grow quickly as the grid dimensions increase. The SAT implementation finds solutions up to grid size $10 \times 10$ and FFF1 until $8 \times 8$.

## 8 Conclusion

We have investigated methods and complexity of testing the termination and confluence of indirect effects expressed as condition-effect rules, showing the main decision problems PSPACE-complete, and giving incomplete but broadly applicable polynomial-time heuristic methods. Additionally, we have presented scalable implementation both for heuristic state space search and logic-based methods, which have

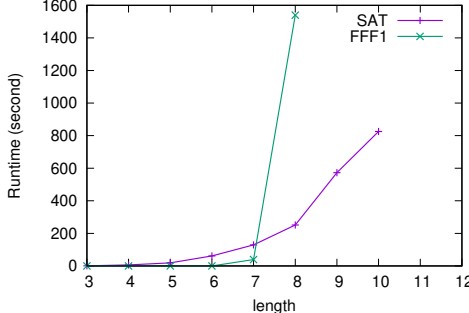

Figure 4: FFF1 and SAT runtimes on Game of Life

not been presented in earlier research. Indirect effects can formalize complex uncontrollable but predictable environments, or they can be understood as control knowledge, as discussed in our experiments.

Future work includes further polynomial-time testable structural properties that guarantee confluence.

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
