# OpenReview forum: "Termination Properties of Transition Rules for Indirect Effects"
_icaps-conference.org/ICAPS/2024/Conference — ICAPS 2024_

### Official Review · Reviewer_ibGX · 2024-01-22

**Significance And Importance:** 2
**Soundness:** 2
**Novelty:** 2
**Clarity:** 3
**Overall Evaluation:** 1
**Confidence:** 3

**Weaknesses:**

1: Minor weaknesses that are easily fixable.

**Contributions Of The Paper:**

The paper concerns "indirect effects" that, in a nutshell, can be triggered whenever their precondition is met. The paper represents indirect effects by "forced actions" that have the same syntax as classical actions, however, they are triggered whenever their precondition is met. The paper then introduces the properties of "termination" and "confluence" meaning whether forced execution (of forced actions) always terminate, and, if so, whether they terminate in the same state (the "confluence" property).

Theoretical findings of the paper are that deciding both properties is PSPACE-complete, and hence the paper identifies under which conditions the confluence can be determined in polynomial time (Theorem 3 and 5). The proposed methods are implemented within the FF planner and a SAT-based planner. The methods are evaluated on a Logistics domain, where forced actions serve as a control knowledge, and on Game of Life, where indirect effects are natural to model the dynamics of the game.

**Ethical Considerations:**

(1) Not Applicable: The paper does not have any ethical considerations to address

**Nomination For Best Paper:**

No

**Questions For Authors:**

1. Is the condition of Theorem 3 a special case of the condition of Theorem 5 ?
2. Would it be possible to extract aggregated effects of the forced actions from an enabling graph ? (i.e., what literals change and how ?)

Post rebuttal:

Thank you for your answers.

I agree that Theorem 3 does not require an enabling graph (and thus its acyclicity). I still feel that a large part of the proof of Theorem 3 overlaps with the proof of Theorem 5, and as forced actions represent indirect effects of an action, the use of enabling graph seems to be natural. On the other hand, I see it now less of an issue that it was before the rebuttal.

If the paper is accepted, I would recommend the authors to add at least a brief paragraph comparing the use of forced actions as Control Knowledge with some well known techniques related to use of Control Knowledge in planning in the literature.

**Reproducibility:**

3: Authors describe the implementation and domains in sufficient detail.

**Strengths Of The Paper:**

Indirect effects, although studied in the literature in the past, have not get much attention recently. The paper shows potential benefits of using those, as shown in Game of Life, indirect effect can be a natural way to model some aspects of the domain, and, as shown in the Logistics domain, they can be exploited as control knowledge to improve performance of planners.

I also appreciate the complexity results (for both termination and confluence properties) and I like the focus on sufficient conditions for determining confluence that can be verified in polynomial time (even though it might identify only subset of cases of confluence).

The paper is well written and the presented concepts are understandable.

**Weaknesses Of The Paper:**

If I understood it correctly, Theorem 5 subsumes Theorem 4 (and 3), as it provides a more general condition. The main finding of Theorems 3 and 5 is that all enabled forced actions have to be applied at some point and that the enabling graph provides partial ordering of the forced actions. For Theorem 3, forced actions can be applied in any order, and the enabling graph in such a case would have edges only from the action a to each enabled forced action. Hence, it seems to be a special case of the conditions in Theorem 5 and in that context, Theorem 3 is a corollary of Theorem 5 (so the proof of Theorem 3 seems to be redundant).

The use of domain control knowledge in planning is studied in the literature (e.g., control rules, planning programs), yet the paper does not reference any of such works. Although I understand that the use of confluent forced actions as a control knowledge is not the main contribution of the paper, it would be great to include at least a short paragraph putting the use of forced actions in the context of some other types of control knowledge.

A small typo in Def.5: ".. a sequence a1,...,an of actionS"

---

> ### Author Rebuttal · Authors · 2024-01-28
>
> Q1: The conditions of Theorem 3 are a special case of the second condition of Theorem 5, as (F × F) ∩ (R_d ∪ R_i ∪ R_c) will be an empty set, and the second condition of Theorem 5 obviously holds for the empty set.
> However, Theorem 3 does not use the graph G_a (or the relation R_e) and hence does not require it to be acyclic. So neither theorem is a special case of the other.
>
> Q2: The enabling graph does not have sufficient information to extract the aggregated effects of the forced actions *exactly*. This graph specifies the possible chains of forced actions that might be executed after performing the action $a$, however, the exact chain depends on the state in which action $a$ is applied. An upper bound (at most which effects) on the possible indirect effects can be obtained from the enabling graph, but not the exact set. This might of course still be useful for some purposes.
>
> Many thanks for the comments!

---

### Official Review · Reviewer_q79r · 2024-01-22

**Significance And Importance:** 2
**Soundness:** 4
**Novelty:** 3
**Clarity:** 4
**Overall Evaluation:** 2
**Confidence:** 3

**Weaknesses:**

1: Minor weaknesses that are easily fixable.

**Contributions Of The Paper:**

The paper provides a complexity analysis of indirect effects, which are formulated as as condition-effect rules that are applied until termination after an agent's choice in action. The paper shows that proving the termination or deterministic outcome of a rule set is PSPACE-complete, equivalent to the standard propositional planning problem. To combat this, the paper introduces syntactic rules which identifies a subset of the rule sets guaranteed to terminate in polynomial time and with the same resulting state regardless of rule order. After proving the correctness of the rules, the paper provides an implementation of the rules in FF and a SATPLAN-style planner, and evaluates the implementations on two domains.

**Ethical Considerations:**

(1) Not Applicable: The paper does not have any ethical considerations to address

**Nomination For Best Paper:**

No

**Questions For Authors:**

1. Will code and experimental data be made available?

2. L396: the second {a} looks redundant and limiting - is there any reason to include interactions back to the non-forced action. Theorem 3 excluded those interactions.

3. What did you mean in line L52 (referenced above)?

======
Post-rebuttal:
I disagree with the authors that choosing older generations of planners has no effect. E.g., one could hypothesize that the advances of the underlying SAT solvers could explain the relative difference in solving time between the FF and SAT planners with and without CK, and that a comparison with with Lama for FD would also show less relative difference than FF.

On the GoL domain, I see less issue - it is good example of how transition rules enable planning in more dynamic (but still deterministic) domains, and its main contribution is as an example of the use of transition rules and showing that the implementation is feasible.

I remain eager to see the work published. I have a number of applications that I've run across that have dynamic portions of the environment, and having language support for transition rules would be helpful.

**Reproducibility:**

3: Authors describe the implementation and domains in sufficient detail.

**Strengths Of The Paper:**

Impact:
Taken together, the paper is a persuasive argument that modern planning languages should include support for indirect effects, as they are practically useful and can be implemented with ease, but need language support in order to be efficiently utilized.

Clarity:
The definitions and proofs are clearly written and organized.

Evaluation:
The evaluation shows that indirect effects can be efficiently integrated into two disparate styles of planners. It also shows applications of indirect effects, both as control knowledge and for implementing complex domain mechanics.

**Weaknesses Of The Paper:**

Experimental evaluation (minor, but almost major):
The planners chosen for the evaluation are more than twenty years old. That makes it difficult to generalize the outcome to modern planners with more powerful heuristics and finely-tuned search implementations. It also limits the uptake of the work if accepted, as the techniques will have to be reimplemented from scratch to integrate into new research and applications.

Unsupported statement in intro (minor):
L52: PSPACE completeness implies no general polynomial time reduction
Either this statement is wrong, since all PSPACE-complete problems can be transformed into each other in polynomial time, or it refers to something else, such as without changing the language of accepted plans, in which case it needs a citation or more justification.

For fun, here is such a translation from a language with forced actions to one without:
Add two new propositions - forced and unforced. Add 'unforced' and 'forced' to the precondition of all unforced and forced actions, respectively. Add \neg unforced and forced to the effects of every unforced action. Add a new action whose precondition is the conjunction of the negation of every unforced action's precondition, and whose effect is to assert unforced and retract forced. Add unforced to the initial state.
To be clear, this translation does not diminish the work of the authors - planners would not be able to identify if and when all sequences of forced actions are confluent, and would likely search fruitlessly through different orders. Better translation might be possible for certain classes of indirect effects, such as those matching theorem 3 or 5, by enforcing an order of applications.


Errata:


L351: Example 3 violates Assumption 3 that forced events make their preconditions false

---

> ### Author Rebuttal · Authors · 2024-01-28
>
> Comment on the implementations: The implementation methods directly apply also to newer planners. In particular, for planners that use heuristic search is easy and requires only a small number of simple modifications. If there is broader interest in implementing this extension to newer planners, the effort to do this is minimal.
>
> Comment on Example 3: The reviewer is right, the example is not compatible with Assumption 3 and should be modified. The simplest example to illustrate the point is (p, {-p, q}), (q,{-q,r}).
>
> Q1: Yes, all code and other data will be made available upon publication.
>
> Q2: The second {a} on line 396 should not be there and looks confusing. Similarly the definitions of R_c, R_e, R_d and R_i unnecessarily include the regular actions in the second component of the relations. Will clean these up.
>
> Q3: "What did you mean in line L52?" The reviewer is right. Yes, (simple) reductions of planning with indirect effects to regular classical planning exist, but these reductions are not practically interesting, and their existence is theoretically obvious. Our sentence, which should have been qualified better, is referring to the possibility of "compiling away" indirect effects, so that each action would include all of its indirect effects as its regular effects. For a given action this compilation is possible only under confluence and termination, so doing this compilation solves these two PSPACE-complete problems and is therefore PSPACE-hard. We should have been clearer on this.

---

### Official Review · Reviewer_FHH9 · 2024-01-22

**Significance And Importance:** 2
**Soundness:** 2
**Novelty:** 2
**Clarity:** 3
**Overall Evaluation:** 1
**Confidence:** 3

**Weaknesses:**

1: Minor weaknesses that are easily fixable.

**Contributions Of The Paper:**

This paper discusses the properties of indirect actions such as confluence and termination and provides new subsets and approximations for discussing confluence for use in planning transition implementations.

**Ethical Considerations:**

(1) Not Applicable: The paper does not have any ethical considerations to address

**Nomination For Best Paper:**

No

**Questions For Authors:**

1) One thing I was thinking about while reading this paper is how the area of static causal laws relate to this papers use of indirect actions, as they have very similar uses with closure functions? If these are completely separate concepts, how does this setup for transition functions differ from those closure function transitions?

2) Can you provide an example of indirect action in a given domain rather than at the abstract levels of (a,not_a)? Put another way, what types of problems or domains would this open up for future work?

**Reproducibility:**

1: Difficult to reproduce because of missing detail.

**Strengths Of The Paper:**

The theories are very consistent in this paper, and the paper builds up everything needed it discusses later on.

**Weaknesses Of The Paper:**

Major: A Major weakness of this paper involves its experimental section, especially its Figures 1 and 2. The points/boxes are unlabeled here and I am not certain what is to be gleaned from these graphs. The paper discusses this being runtime but is the runtime the x-axis?   The x-axis is just labeled as "base".  Either a table breaking down the differences or something more akin to Figure 4 would greatly strengthen the argument/case for what is being discussed around line 670.

Minor: Overall, this paper is very... high-level. I found it difficult to follow along in the abstract when no example domain nor case study was given to help root the problem space to some form of specific forced-action. I was hoping the introduction of the logistics domain and game of life at the end would help, but they come off as slightly arbitrary. (Some of this is followed up with in the Questions section)

---

> ### Author Rebuttal · Authors · 2024-01-28
>
> Figures 1 & 2: These are standard scatter plots, with FFF1 (Forced Action) variant runtime (in seconds) plotted against FF (Base) runtimes in Figure 1, and similarly for the SAT implementation in Figure 2. Each small square represents one instance, and one can read the respective runtimes from the Y-axis (for Forced Actions) and from the X-axis (for the Base case).
>
> Q1: The concept of static causal laws is closely related to indirect actions; As mentioned in the related work, Planning research has adopted different notations for this concept (which, we believe, demonstrates the importance of the topic in question).
> Closure function, as defined by M. Gelfond & R. Watson (1999), is a function that returns the *least superset* that is closed under the static causal law, while, in our paper, we investigate the sufficient confluent conditions and *uniqueness* of the terminal state.
> Much of work on "causal laws" in the Reasoning about Actions community has assumed that the static laws are non-conflicting (the set of their effects is consistent). We believe this has been to avoid complications like non-confluence and the need to thoroughly look at the different possible execution orders of the causal laws.
>
> Gelfond, M., & Watson, R. (1999). On Methodology of Representing Knowledge in Dynamic Domains. Electronic Notes in Theoretical Computer Science, 25, 121-132.
>
> Q2: Domains that have motivated our work are found in software applications. For example in an information system for banking, an indirect effect of tranferring money to an account could be the the payment of bills that were set to be automatically paid but that could not be paid before due to insufficient funds in the account. Similar indirect effects of users' actions are found in almost all slightly more complex software systems.
> The examples in the paper include the Game of Life domain, presented in the experiments section. In this example, the state of each cell, whether it is alive or dead, is determined by the rules of the environment; these rules can conveniently be described by indirect effects while it is impractical to express them by the standard modeling languages, and they are also not naturally part of the user's actions.

---

### Meta-Review · Area_Chair_BNEo · 2024-02-04

**Recommendation:** Accept (Oral)
**Confidence:** 5

**Metareview:**

The reviewers agree that the paper makes a compelling argument for using indirect effects in planning languages. The theory is presented clearly and provides a thorough analysis of the topic (including complexity results), and the experimental analysis convicingly showcases the advantages of the approach. While the paper should be accepted, the discussion brought up two poins which the authors must integrate in the final version of the paper:

1) Clarify Figures 1 and 2 by briefly explaining in the text what the plots show (see review FHH9).
2) Add a brief discussion of related work on the use of Control Knowledge (see post rebuttal comment in review ibGX).

**Ethical Considerations:**

(1) Not Applicable: The paper does not have any ethical considerations to address